# The Diverse Salt-Stress Response of Arabidopsis *ctr1-1* and *ein2-1* Ethylene Signaling Mutants Is Linked to Altered Root Auxin Homeostasis

**DOI:** 10.3390/plants10030452

**Published:** 2021-02-27

**Authors:** Irina I. Vaseva, Kiril Mishev, Thomas Depaepe, Valya Vassileva, Dominique Van Der Straeten

**Affiliations:** 1Department of Molecular Biology and Genetics, Institute of Plant Physiology and Genetics, Bulgarian Academy of Sciences, Acad. Georgi Bonchev Str., Bldg. 21, 1113 Sofia, Bulgaria; mishev@bio21.bas.bg (K.M.); valyavassileva@bio21.bas.bg (V.V.); 2Laboratory of Functional Plant Biology, Department of Biology, Ghent University, K.L. Ledeganckststraat 35, B-9000 Ghent, Belgium; thdpaepe.Depaepe@UGent.be (T.D.); Dominique.VanDerStraeten@UGent.be (D.V.D.S.)

**Keywords:** Arabidopsis *ctr1-1* and *ein2-1* mutants, auxin homeostasis, ethylene signals, crosstalk, salt stress

## Abstract

We explored the interplay between ethylene signals and the auxin pool in roots exposed to high salinity using *Arabidopsis*
*thaliana* wild-type plants (Col-0), and the ethylene-signaling mutants *ctr1-1* (constitutive) and *ein2-1* (insensitive). The negative effect of salt stress was less pronounced in *ctr1-1* individuals, which was concomitant with augmented auxin signaling both in the *ctr1-1* controls and after 100 mM NaCl treatment. The R2D2 auxin sensorallowed mapping this active auxin increase to the root epidermal cells in the late Cell Division (CDZ) and Transition Zone (TZ). In contrast, the ethylene-insensitive *ein2-1* plants appeared depleted in active auxins. The involvement of ethylene/auxin crosstalk in the salt stress response was evaluated by introducing auxin reporters for local biosynthesis (*pTAR2::GUS*) and polar transport (*pLAX3::GUS*, *pAUX1::AUX1-YFP*, *pPIN1::PIN1-GFP*, *pPIN2::PIN2-GFP*, *pPIN3::GUS*) in the mutants. The constantly operating ethylene-signaling pathway in *ctr1-1* was linked to increased auxin biosynthesis. This was accompanied by a steady expression of the auxin transporters evaluated by qRT-PCR and crosses with the auxin transport reporters. The results imply that the ability of *ctr1-1* mutant to tolerate high salinity could be related to the altered ethylene/auxin regulatory loop manifested by a stabilized local auxin biosynthesis and transport.

## 1. Introduction

Saline soils present a serious agricultural constraint especially in coastal areas and in regions with industrial pollution or intensive plant breeding. Moderate salt stress could remain undetected since it causes no apparent injuries other than restricted growth. This means that high salinity has a negative effect on the signaling cascades involved in the regulation of plant growth and development. The growth restriction caused by high salt concentrations is further complicated by impaired photosynthesis which ultimately leads to accelerated aging and death [1,2].

Plants adapt to environmental challenges through anatomical, metabolic, and morphological changes and the gaseous plant hormone ethylene modulates many of these growth-related processes. As a major stress hormone, ethylene causes growth reduction primarily due to the inhibition of cell expansion which is an adaptive response to the adverse environment [3,4]. Several studies have demonstrated that plants exposed to salt stress show induced ethylene biosynthesis and have enhanced ethylene signaling maintaining both shoot and primary root growth [5,6,7,8,9,10]. Some ethylene mutations, which are characterized with increased ethylene biosynthesis, tend to absorb less Na from the soil that ultimately results in improved shoot Na/K homeostasis and thus enhanced salt tolerance [9]. The epidermis has been identified as the main site of ethylene-controlled growth inhibition [11]. The root epidermis in particular, being in direct contact with the soil substrate, probably is of utmost importance in the growth arrest response under adverse environmental conditions. The existing experimental evidence for organ-specific regulatory mechanisms in response to salt stress [12] calls for the elucidation of ethylene signal input at an organ-specific basis.

In the absence of the hormone, the ethylene-signaling cascade is suppressed via a Raf-like kinase Constitutive Triple Response 1(CTR1) which inhibits the endoplasmic reticulum-localized Ethylene Insensitive 2 (EIN2) by direct phosphorylation [13,14]. Upon binding to the receptors, ethylene inactivates CTR1 releasing the repression of EIN2 which through its C-terminus activates the master regulators of ethylene response, Ethylene Insensitive 3 (EIN3) and EIN3-Like 1 (EIL1) in the nucleus [15,16]. Mechanisms conferring salinity tolerance by deterring Reactive Oxygen Species ROS accumulation have been linked to salt-induced stabilization of EIN3 and EIL1 [17]. However, salt tolerance acquisition would integrate a number of factors which are controlled by complex regulatory mechanisms.

The present work proposes some insights on the implication of ethylene-signaling through CTR1 and EIN2 in ethylene/auxin crosstalk which affects auxin homeostasis under salt stress. We put a particular focus on the ethylene-related regulation of auxin accumulation, by exploring root local auxin biosynthesis and transport. The changes in active auxins in the mutants *ethylene insensitive 2-1* (*ein2-1)* and *constitutive triple response 1-1* (*ctr1-1)* under high salinity stress were monitored through the classical *pDR5::GUS* auxin reporter [18] and the highly sensitive auxin sensor R2D2 (“Ratiometric version of 2 D2’s”) [19] in homozygous (F3) individuals. Ethylene-insensitive *ein2* mutants have been independently isolated in genetic screens designed to identify components of other hormone signaling pathways [20,21,22,23]. This anticipates that EIN2 acts as a hub for multiple plant hormones outputs involved in stress responses and it presents a suitable target for evaluation of changes in stress-tolerance. For example it has been shown that EIN2 regulates plant response to osmotic and salt stress through an ABA-dependent pathway in Arabidopsis [7]. Additionally, the downstream Ethylene Response Factors (ERFs) are pivotal in linking hormone signaling and stress tolerance [24].

Transport-driven auxin accumulation, resulting in local auxin maxima, is crucial for the plant response to gravity and it is a key process in the asymmetric root growth provoked by environmental and developmental stimuli [25,26]. It has been recently established that local auxin minima present a signal triggering the transition from cell division to cell differentiation [27], a process which is also closely implicated in the adaptive responses towards unfavorable environment. Nevertheless, the importance of the changes in polar auxin transport during abiotic stresses remains insufficiently explored [28].

Differential organ growth across gravity vector (roots and stems) is provoked by external stressors like penetration through dense substrates or avoidance of unfavorable environment. This phenomenon is generally assigned to the differential lateral auxin transport [29,30] for which there is substantial evidence of its coordinated regulation by ethylene signals [11,31,32]. Salt also modulates root growth direction by reducing the gravity response [33]. Besides, it has been demonstrated that increased ethylene concentration reduces gravitropic curvature [34] and it also suppresses the root-bending reaction to physical hardness [35].

There is limited information on ethylene/auxin interplay controlling the differential root elongation, or halotropism on high-salinity substrates. It has been demonstrated that the directionality of this response involves PIN-FORMED 2 (PIN2) auxin efflux carrier which participates in the auxin redistribution in the root tip [36].

Here, the ethylene signals affecting auxin polar transport in adaptive root growth under salt stress were addressed by monitoring the expression of auxin transporters in *ctr1-1* and *ein2-1* background via quantitative reverse transcription PCR (qRT-PCR) analyses and through genetic crosses with auxin reporter lines. The major criterion for the selection of the auxin reporter lines included in the study was their activation by 1-aminocyclopropane-1-carboxylic acid (ACC, the precursor of ethylene) attesting to the involvement of ethylene signaling in the control mechanisms [37] and the presence of AP2/ERBP binding sites in the gene promoters identified by AthaMap analyses [38]. The cell-type specificity of the reporter’s expression was also considered in terms of sufficient coverage of the different cell files in the root tip zones. We evaluated the changes of the expression in the efflux carriers *PIN-FORMED1 (PIN1), PIN2,* and *PIN-FORMED3 (PIN3)* as well as the influx carriers *AUXIN RESISTANT 1* (*AUX1*) and *LIKE AUX1 3* (*LAX3*) under high salinity, as it has been already demonstrated that some of them were involved in auxin–ethylene interactions [31].

## 2. Results

### 2.1. Altered Gravity Bending of the Ethylene-Signaling Mutants at High Salinity

Gravitropism in shoots and roots results from asymmetric growth involving cooperative hormone action with auxins operating as the main mediators [39]. Disturbed root bending upon gravitropic stimulation is indicative of defects in auxin-dependent processes such as vesicle trafficking and cytoskeletal organization [40,41].The root elongation and gravitropic response on high salinity substrate outlined the differential reaction of the constitutive ethylene-signaling mutant *ctr1-1* to salt stress compared to the wild type and *ein2-1* (Figure 1a).

We measured the root response angle of gravistimulated control individuals grown on half-strength Murashige and Scoog medium (½ MS,) and others grown on ½ MS supplemented with 100 mM NaCl (salt-treated) (Figure 1b). The gravitropic responses of the tested genotypes even under control conditions displayed the differential behavior of the ethylene mutants compared to the wild type (Figure 1a,b). Both the wild type and *ein2-1* plants exhibited impaired gravitropic bending under salinity stress, with the majority of seedlings showing an average angle between 120° and 150° (Figure 1b). This was observed in 72% of the wild-type individuals and in 54% of the mutant *ein2-1* plants grown in the presence of 100 mM NaCl. The more pronounced loss of gravitropic response in *ein2-1* compared to the wild type is likely linked to disturbed auxin homeostasis leading to a malfunctioning reaction to gravitropic stimuli. In contrast, the ethylene constitutive mutant *ctr1-1* showed no significant alterations in gravitropic bending upon salt stress (Figure 1a,b).

The root elongation of the three tested genotypes in the same assay was negatively affected by high salt concentrations but this effect was stronger in both the wild type and the *ein2-1* plants (approximately 60% of the control) and less pronounced in *ctr1-1* individuals (80% of the control nontreated seedlings) (Figure 1c).

The evaluation of the gravitropic bending response and root elongation under salt stress confirmed that the mutant genotypes performed differently compared to the wild type. These findings incite exploration of auxin-related processes controlled by ethylene signals that might be involved in the directional root growth (halotropic response) to avoid high salt concentrations.

### 2.2. Differential Performance of the Ethylene-Signaling Mutants under Salt Stress Corresponds to Severely Altered Auxin Content in the Primary Root

Two different auxin reporting systems were used to monitor the auxin changes in response to salt stress. We introduced the classical auxin reporter *pDR5::GUS* [18] in *ctr1-1* and *ein2-1* mutants via genetic crosses and analyzed homozygous F3 individuals subjected to 100 mM NaCl for 48 h (Figure 2a). The short incubation in staining solution (1 h) revealed inhibition of DR5 signal in the root meristem region of the salt-treated wild type individuals (Figure 2a) confirming previously published results with *pDR5::GFP* reporter in a study on the negative effect of salt stress on the root meristem size [42].

Notably, the DR5 signal in the salt-stressed *ctr1-1* plants did not show any reduction of the signal after one hour of GUS staining but instead, it spread in the Lateral Root Cap (LRC) and epidermal cells. This pattern was even more pronounced, in both *ctr1-1* controls and salt-treated seedlings after prolonged incubation in GUS staining solution (for two and three hours). This suggests that the salt treatment did not significantly change the DR5 reporter expression in the *ctr1-1* mutant (Figure 2a). However, as reported by the *pDR5::GUS* construct introduced in *ein2-1* plants, even after longer staining it appeared that the active auxins in both control and stressed roots of the ethylene-insensitive mutant remained distinctively depleted (Figure 2a,b).

The quantification of whole frame GUS staining intensity, which includes the intact Cell Division Zone (CDZ) and Transition Zone (TZ), in individuals stained for 2 h showed that the NaCl treatment slightly but significantly increased the signal in the reporter roots (Col-0 background). This effect was not observed when the intensity measurements were taken only in the meristem region, where the prolonged staining has caused oversaturated signal (Figure 2b). Therefore the detected salt-stress-related increase in auxin signaling after 2 h of GUS staining should be assigned mostly to the cell types outside this part of the root tip.

The assumption that the increased auxin activity provoked by the salt stress should be assigned to the cells outside the root meristem was confirmed by the measurement of the fluorescent signal in the NaCl-treated highly sensitive auxin sensor R2D2 (Figure 2c). The latter is based on the auxin-dependent degradation domain II (DII) of Aux/IAA proteins and combines *RPS5A*-driven DII fused to n3 × Venus and *RPS5A*-driven mutated-DII fused to ndtTomato on a single transgene [19]. Such a ratiometric version of two DIIs allows high-definition evaluation of changes in active auxin levels in distinct cell files. The R2D2 auxin sensor revealed a cell type-specific response to 100 mM NaCl in the epidermal cell layer of the root tip. The mere fact that this is not observed when the sensor operates in *ein2-1* ethylene-insensitive genetic background, links this particular auxin salt stress response to ethylene signaling (Figure 2c). The low green fluorescent signal both in the salt-stressed and in the control homozygous F3 *ctr1-1* individuals carrying R2D2 reporter demonstrated that the constitutive mutant is characterized with higher basal auxin content in the primary root which remained relatively stable after 48 h of 100 mM NaCl treatment.

### 2.3. Constitutive Mutant ctr1-1 Has Increased Local Auxin Biosynthesis

Indole-3-acetic acid (IAA) is synthesized through multiple biosynthetic pathways, and the indole pyruvate (IPA) pathway which utilizes the amino acid tryptophan (Trp) as auxin precursor has been identified as a major route. The first steps in auxin biosynthesis are catalyzed by a protein family represented by Tryptophan Aminotransferase of Arabidopsis 1 (TAA1), Tryptophan Aminotransferase Related 1 (TAR1) and Tryptophan Aminotransferase Related 2 (TAR2). TAA1, also known as Weak Ethylene Insensitive 8 (WEI8), has been initially identified from a mutant screen for weak ethylene insensitivity of root growth. [43]. These enzymes convert tryptophan to IPA which is further converted to the active auxin IAA by YUCCA (YUC) proteins, a family of flavin-dependent monooxygenases [44]. Since the changes in local auxin biosynthesis affect root responses during abiotic stress [45,46], the involvement of ethylene signals in the process by initial in silico characterization of the genes coding for the enzymes from the Trp biosynthetic pathway has been checked. We also assessed the promoter activity of *TAR2* in the *ctr1-1* and *ein2-1* mutant backgrounds by introducing the *pTAR2::GUS* reporter via genetic crosses.

The performed AthaMap analysis [38] of the promoter regions of the genes encoding TAA1, TAR1, and TAR2 identified the consistent presence of motifs for APETALA2/Ethylene-responsive element binding proteins (AP2/ERBP), among which were Ethylene Response Factor 1 (ERF1), Ethylene Response Factor 2 (ERF2), Ethylene-Insensitive3-Like (TEIL), Target of Early Activation Tagged 1 /2 (TOE1/2), and EIN3 (Appendix A). The ethylene regulatory link to local auxin biosynthesis was also assessed by the TF DEACoN (Transcription Factor Discovery by Enrichment Analysis of Co-expression Networks) tool [47] which uses Arabidopsis DNA affinity purification sequencing (DAP-Seq) data to make predictions about which TFs may be involved in transcriptional responses of co-regulated genes.

TF DEACoN analysis of the genes encoding enzymes from the Tryptophan-dependent auxin biosynthesis (TAA1, TAR1, TAR2, YUC1-11) identified three AP2-EREBP transcription factors (RAP2.1, RAP2.9, and TINY or also known as ERF040) with fourfold enrichment of targets (Appendix A). Taken together with previously published results [11,46] these in silico evaluation confirms the involvement of ethylene signaling in the regulation of local auxin biosynthesis [48].

TAR2 participates in the ethylene-directed IAA biosynthesis [49] and it has been linked to ethylene-signaling pathway through YUC8 [50,51]. As shown in Figure 3, the ethylene precursor ACC was able to induce expression of *pTAR2::GUS* in the primary root, undetectable under control conditions. We introduced the *pTAR2::GUS* reporter in *ein2-1* and *ctr1-1* mutants and evaluated its expression in homozygous F3 individuals grown in the presence of 100 mM NaCl for 48 h (Figure 3). The GUS staining pattern revealed that under normal conditions TAR2 is detected mainly in the developing leaves of the wild type, and that the exposure to NaCl suppressed its expression. The signal was undetectable in the ethylene-insensitive *ein2-1* background both in untreated and in salt-stressed individuals. However, strong *pTAR2::GUS* staining both in the young leaves and in the roots was observed in the nonstressed and salt-treated plants that carry the *ctr1-1* mutation. This means that the constitutive ethylene-signaling mutant is characterized with ethylene-induced strong local auxin biosynthesis which remains slightly affected by the stress treatment.

### 2.4. Stabilized Polar AuxinTtransport in Salt-Treated ctr1-1 Plants

Polar auxin transport is an important factor contributing to the dynamic changes of the hormone in response to environmental and developmental stimuli and it has been demonstrated that the expression of several auxin transporters depend on ethylene signaling [11,32,52].

#### 2.4.1. In Silico Characterization of Auxin Transport Genes Reveals Regulation by Ethylene-Related Transcription Factors

AthaMap [38] (Appendix A) and TF DEACoN analyses [47] (Appendix A) of the auxin transport genes were initially performed to check for ethylene-related regulatory elements. AthaMap analysis of the promoter regions of the auxin transporter genes *AUX1, LAX3, PIN1, PIN2* and *PIN3* outlined the presence of a number of AP2/ERBP transcription factors binding sites, including AtERF1 binding sequences in *ProLAX3* and *ProPIN1* as well as indications for the involvement of other AP2-EREBP transcription factors (RAP2.6 also known as ERF113, TEIL, DEAR3, ERF2, TOE1, TOE3, RAV2) (Appendix A).

TF DEACoN analysis of the known auxin transporter genes (*PIN1-8*, *ABCB*-type transporters, as well *AUX1* and *LAX1/2/3*) returned three AP2-EREBP type transcription factors (RAP2.1, ERF014, and TINY) with fourfold enrichment of targets (Appendix A).

#### 2.4.2. Expression Profiling of Auxin Transporter Genes

The expression profiling of auxin transporter genes in root-derived samples from plants grown on media containing 100 mM NaCl for 14 consecutive days (14 DAG), outlined the genes encoding influx carrier *AUX1*, as well as the efflux carrier *PIN1* as significantly affected by the *ctr1-1* mutation (Figure 4a,b).

Overall, the long-term cultivation on media containing 100 mM NaCl was linked to a slightly increased expression of influx (*AUX1* and *LAX3*) and efflux (*PIN1* and *PIN2*) carrier genes in the wild type but this change was not statistically significant. However, the expression of these genes in the *ctr1-1* salt-stressed roots (except *PIN2*) was significantly higher than in the untreated *ctr1-1* controls and it increased by two (*PIN1*) to four fold (*AUX1*) compared to the levels detected in the wild type and ethylene-insensitive *ein2-1* mutant plants. The other gene *LAX3* exhibited higher relative expression in the *ctr1-1* mutants compared to the wild type and *ein2-1* under salt stress conditions but the difference was not statistically significant (Figure 4a).

#### 2.4.3. Analyses of Auxin Transport Reporters in Ethylene Mutant Backgrounds

To address the effect of the ethylene signals on polar auxin transport in the roots subjected to salt stress with a cell-type-specific resolution we introduced auxin transport reporter constructs (*pLAX3::GUS, pAUX1::AUX1-YFP, pPIN1::PIN1-GFP*, *pPIN2::PIN2-GFP* and *pPIN3::GUS*) into *ctr1-1* and *ein2-1* by crossing. The analyses were performed with seven-day-old homozygous (F3) plants subjected to 48 h cultivation on media containing 100 mM NaCl. The expression of *pLAX3::GUS* reporter was not significantly affected by the salt stress in all the tested lines (Figure 5a) but the overall *pLAX3::GUS* staining intensity in *ctr1-1* background appeared stronger than the one detected in wild type and in the ethylene-insensitive mutant *ein2-1* under both control and salt stress conditions (Figure 5a). This observation is in line with the measured stronger salt-induced *LAX3* expression in *ctr1-1* roots detected by qRT-PCR (Figure 4a).

We found that salt stress inhibited the fluorescent signal in the root stele of the *pAUX1::AUX1-YFP* reporter line without having a significant effect on the intensity in the epidermal cells (Figure 5b). The expression of the reporter in ethylene-insensitive *ein2-1* mutants followed the pattern observed in Col-0. It remained relatively unchanged in the epidermal cell files after NaCl application but it was reduced in the stele of the salt-stressed individuals.

The overall intensity of the reporter in *ctr1-1* background under control and salt-stress conditions was found to be relatively lower than the one measured in the wild type and *ein2-1* plants (Figure 5b). As opposed to the wild type and the ethylene-insensitive mutant, the salt stress induced *pAUX1::AUX1-YFP* expression in the epidermal cells in *ctr1-1* individuals without exhibiting any significant negative effect over the fluorescent signal in the stele (Figure 5b). These observations suggest that the ethylene signaling has a cell type-specific effect on the auxin import via AUX1 under salt stress and it is mainly assigned to the root tip epidermis.

We evaluated the effect of the impaired ethylene signals on auxin export by introducing efflux carrier reporters in the constitutive and the insensitive mutants. The expression of the *pPIN3::GUS* reporter in the wild type Col-0 showed an increase in the salt-stressed individuals (Figure 6a). The signal was not detected in the root tips of the ethylene-insensitive mutants. However, *pPIN3::GUS* staining was found to be stable and strong in *ctr1-1* plants 48 h after they were transferred on media containing 100 mM NaCl. An interesting detail on the expression pattern in the *ctr1-1* background was documented: the *pPIN3::GUS* staining in the root apical meristem was not detected in the constitutive mutant but it appeared stronger in the vascular cell files (Figure 6a). This observation accounts for a probable differential involvement of ethylene/auxin crosstalk depending on the cell type context. The accumulation of *PIN3* transcripts in the samples derived from whole roots seemed independent of the ethylene mutations (Figure 4b). The varying *PIN3* expression profile in the root tip identified by GUS staining, particularly the low staining of the homozygous *ein2-1*, further suggests that the involvement of ethylene signals in the control of root auxin balance should be addressed in a cell-type-specific manner. Salt stress reduced the fluorescent signal in the *pPIN1::PIN1-GFP* reporter line confirming the earlier observation that high salinity inhibits the auxin efflux *PIN1* expression [42] (Figure 6b). When introduced into the ethylene-insensitive mutant *ein2-1* the same trend was observed although the overall intensity of the signal was lower than the one observed in the wild type genetic background. We observed strong *pPIN1::PIN1-GFP* expression in *ctr1-1* mutant background which seemed not affected by the applied salt stress (Figure 6b). This means that PIN1 carrier, which is expressed mainly in the root stele and endodermis cells, is positively regulated by ethylene signals.

The operating in the root epidermal cells exporter PIN2 directs the auxin flow from the tip into the root elongation zone (basipetal auxin transport). This efflux carrier was previously found to be involved in the auxin redistribution in the root tip triggering directional growth response provoked by higher salinity of the substrate [36]. The effect of the 48 h treatment with 100 mM NaCl did not affect significantly the fluorescent signal of the *pPIN2::PIN2-GFP* reporter in Col-0 and in *ein2-1* background (Figure 6c). However, we found that the expression of the reporter in the constitutive ethylene-signaling mutant *ctr1-1* under control conditions was notably stronger than the one detected in the wild type and in the insensitive mutant. This suggests a positive regulation of the PIN2 transporter by ethylene signals which is in line with previously published results with this reporter [11]. The applied salt stress provoked inhibition of the fluorescent signal reaching levels similar to the ones detected in the wild type and the *ein2-1* backgrounds (Figure 6c).

The observed reciprocal effect of the constitutive ethylene-signaling mutation over the acropetal (rootward) auxin transport through the efflux carriers PIN1 and PIN3 in the vascular tissue, and the shootward (basipetal) transport through PIN2 in the epidermis further strengthens the cell-type-specific aspects of the crosstalk between the two hormones regulating both the directional root growth, and root growth inhibition under salt stress.

## 3. Discussion

Understanding the gene regulatory network of the salt stress response will provide useful information for molecular-based strategies aiming to improve crop performance on high salinity soils. A substantial part of this knowledge is related to the identification of interconnections among various signaling cascades and the processes that they trigger. Auxin-mediated regulation of growth and development during abiotic stress, including salt stress, is determined by the changes in auxin transport, biosynthesis, conjugation, perception, and signaling [26,28]. The strong correlation between auxin content and stress biology has been convincingly demonstrated showing that perturbations in the hormone levels cause antioxidant accumulation and stress responses in Arabidopsis [53]. A number of previous studies have also revealed the link between adaptation responses to high salinity and auxin physiology [42,54,55,56].

Root elongation is to a large extent controlled by auxins and their synergistic or antagonistic interplay with ethylene which regulates growth through effects on auxin biosynthesis and transport [11,46,52,57]. It has been also shown that auxins and ethylene promote root hair elongation [58,59,60,61] while their interaction reduces primary root growth [57] and lateral root initiation [62]. Intensified hairy root development increases root surface area facilitating water and nutrient uptake which are beneficial under salt stress conditions [63]. The *ctr1-1* mutant has a constantly activated ethylene signaling and it is characterized by reduced primary root elongation and formation of excessive number of ectopic root hairs pointing at amended basal auxin levels. The increased auxin content in *ctr1-1* plants could be a predisposition for their better performance on challenging substrates, including ones with high salinity. These make the *ctr1-1* mutant a suitable target for evaluation of the ethylene/auxin crosstalk under high salinity stress in comparative experiments with the ethylene-insensitive *ein2-1* in an attempt to provide a mechanistic understanding on how these hormones coordinate their action in the adaptive response. The presented study on the involvement of CTR1 and EIN2 ethylene-signaling components in salt stress responses using mutant *ctr1-1* and *ein2-1* plants has pointed at some of the possible hormonal crosstalk regulatory mechanisms participating in salt tolerance acquisition. Auxin homeostasis is closely involved in the halotropic reactions which are triggered by auxin asymmetric distribution resulting in organ growth directional changes [33,36,64]. Here we provide direct evidence that ethylene signals (*ctr1-1*) positively regulate the activity of important players in the polar auxin transport thus contributing to root directional growth under salt stress. This is further confirmed by the detected low auxin levels and decreased influx/efflux carrier expression in *ein2-1* roots which also explain the impaired gravitropic response of the mutant when grown on media with higher salinity.

It is now well established that local auxin biosynthesis is required for maintaining functional root meristems and that this process has a rather versatile character thus enabling swift response to a plethora of changing environmental factors, ensuring growth plasticity [46,51]. A previous study in maize has reported that the increased auxin content in roots after osmotic stress resulted from a higher rate of IAA biosynthesis [65]. Our results revealed induction of *TAR2* expression in the root tip of *ctr1-1* plants which suggests that ethylene signals might be involved in the control of stress-induced local auxin biosynthesis but this process needs further confirmation.

As pointed earlier by Kaya et al. [1] and Ma et al. [66] the sometimes contradictory results on the effect of high salinity on the auxin pool in different crops, call for a proper evaluation of the involvement of different auxin-triggered physiological processes in this stress response.

The salt-mediated inhibition of root growth in Arabidopsis has been linked to inhibited auxin signaling and reduced expression of auxin PIN genes [42]. The results in this study, summarized in the scheme presented in Figure 7, demonstrated that salt stress has a differential effect on the polar auxin transport in the stele (through AUX1 and PIN1), root meristem (through PIN3) and epidermis (through PIN2) of the wild type (Col-0), the insensitive *ein2-1* and the *ctr1-1* constitutive ethylene-signaling mutant.

The relatively stable expression of *LAX3* and *PIN3* in the early elongation zone of the wild type plants under salt stress was accompanied by reduced auxin influx/efflux (*AUX1*/*PIN1*) in the CDZ.

Meanwhile, we detected reduced auxin influx (*AUX1*) and slightly intensified efflux (*PIN3*) in the root apical meristem. These versatile polar transport patterns coincide with the observed accumulation of auxins in the epidermis of the TZ/EZ evidenced by the R2D2 salt-stressed individuals. The obstructed import and export of the hormone in the epidermal cells results in abnormal growth-inhibiting auxin levels. When the same salt stress/auxin transport relations were monitored in the *ctr1-1* mutant background, i.e. under constantly operating ethylene signaling, the negative effect of the high salt concentration on the root auxin efflux/influx in the stele were mitigated and the *PIN3* activity in the root meristem region seemed repressed. Moreover, the local auxin biosynthesis in the constitutive mutant appeared unusually increased which could also contribute to its changed auxin dose response, reaching supraoptimal auxin concentrations followed by a rise in ethylene signaling [11] thus explaining the severely affected root elongation growth. When combined with the relatively stable polar transport, the high local auxin content adds yet another element that could explain the better performance of *ctr1-1* plants which is in line with the earlier suggested importance of the stress-triggered local auxin biosynthesis [65].

Our results suggest that ethylene is involved in salt stress response, not only through activation of stress-related molecular cascades, but also through the regulation of auxin biosynthesis and polar transport, both affecting the local auxin balance that shapes the adaptive root growth.

## 4. Materials and Methods

### 4.1. Plant Material and Growth Conditions

*Arabidopsis thaliana* wild type (Col-0) and the ethylene-insensitive mutant *ein2-1* were purchased from the Nottingham Arabidopsis Stock Center—NASC (http://arabidopsis.info/, UK). The constitutive mutant *ctr1-1* was obtained from the Arabidopsis Biological Resource Center, The Ohio State University (https://abrc.osu.edu/, Columbus, Ohio, USA). R2D2 auxin sensor was received from Prof. Dolf Weijers (Wageningen University). The lines *pPIN1::PIN1-GFP* [67], *pPIN2::PIN2-GFP* [68], *pPIN3::GUS* [30], *pAUX1::AUX1-YFP (aux1-22)* [69], *pLAX3::GUS* [70], and *pTAR2::GUS* [43] have been previously described.

*Arabidopsis thaliana* seeds were surface sterilized, and stratified for two days in the dark at 4 °C. After that the plants were grown vertically in square plates on half-strength Murashige and Skoog (½ MS) medium (pH 5.7) solidified with 1.0% agar at 22 °C, 16 h light/ 8 h dark cycle and light intensity of ~150 μmol m^−2^ s^−1^. The salt stress was imposed by adding 100 mM NaCl to the media as this was identified as a threshold concentration at which the three analyzed genetic backgrounds (Col-0, *ein2-1* and *ctr1-1*) had comparable germination and survival rates (unpublished results).

The gravitropic bending assay was performed with vertically cultured four-day-old seedlings transferred onto plates with fresh ½ MS medium with or without 100 mM NaCl (no sucrose added). The root elongation and bending angle were evaluated 48 h later. Measurements of three datasets (i.e. three independent experiments, each containing at least 20 individuals per treatment and per genotype) were performed with ImageJ 1.52r.

### 4.2. GUS Staining and Quantification

The four-day-old plants used in the GUS staining assays and in the confocal observations (parental reporter lines and their F3 crosses with the mutants) were analyzed after being transferred on media containing 100 mM NaCl and grown for additional 48 h.

The seedlings were incubated in 90% acetone overnight at 4 °C. Plants were washed twice in 50 mM sodium phosphate buffer (pH 7.2) and then were incubated in 5-bromo-4-chloro-3-indolyl-β-glucuronide (X-gluc) buffer (50 mM NaPO4 (pH 7.2), 0.1% Triton X-100, 4 mM K3(Fe(CN)600), and 2 mM X-gluc) at 37 °C for 1, 2 and 3 h (*pDR5::GUS*), 4 h (*pTAR2::GUS* and *pLAX3::GUS*) or 16 h (*pPIN3::GUS*). The GUS staining was observed under DIC microscope Olympus BX 51 equipped with XC50 digital camera.

The evaluation of mean grey intensity in the red channel of the microscopic images and the background reading was performed in ImageJ 1.52r by using macros created for the specific region of interest (ROI). The results are presented by subtracting the optical density (OD) of the background from OD of the specimen. The graphs show the mean GUS intensity (in arbitrary units) measured within a specified rectangular ROI in the root cell division zone. At least 10 different plants from each experimental group were measured to produce an average value with standard deviation (SD).

### 4.3. Fluorescent and Confocal Microscopic Observations

Evaluation of the changes in green fluorescent signal was done using at least 10 parental (*pPIN1::PIN1-GFP*, *pPIN2::PIN2-GFP*, *pAUX1::AUX1-YFP* and R2D2) and 10 homozygous F3 five-day-old seedlings from the *ein2-1* and *ctr1-1* genetic crosses. The individuals from the different lines were usually grown on one and the same plate to omit variations in growth conditions. Subsequently, in each experiment, the observations were made under the same settings to allow comparison of fluorescent output. The observations of the crosses with *pPIN1::PIN1-GFP*, *pPIN2::PIN2-GFP*, *pAUX1::AUX1-YFP* reporters were made with Olympus BX53 Digital Upright Microscope equipped with CellSense Dimension software (Olympus Life Science, Tokyo, Japan). For R2D2 imaging, whole seedlings were mounted in NaCl-containing ½ MS medium, and images were taken with an Andor Dragonfly spinning disk confocal system (40X water immersion objective lens, NA 1.25) equipped with Fusion software (Andor Technologies, Inc., Belfast, UK) and iXon897 EMCCD camera. The excitation wavelength was 488 nm for GFP, and 561 nm for tdTomato. For quantification, images were converted to 8bit with ImageJ 1.50f. Regions of interest (ROIs) were selected to comprise the epidermal cell layer or the entire CDZ and TZ. Histograms listing all fluorescence intensity values per ROI were generated and the averages of the 100 most intense pixels in the red and green channels were used to calculate the mDII/DII fluorescence intensity ratio.

### 4.4. TF DEACoN and AthaMap Analyses

The TF DEACoN tool (Transcription Factor Discovery by Enrichment Analysis of Co-expression Networks) [47] was used to characterize the TFs which may be involved in the transcriptional response of genes encoding enzymes from the Trp-dependent auxin biosynthesis (*TAA1 – At1g70560*; *TAR1 – At1g23320*; *TAR2 – At4g24670*, and *YUC1/2/3/4/5/6/7/8/9/10/11*: *At4g32540, At4g13260, At1g04610, At5g11320, At5g43890, At5g25620, At2g33230, At4g28720, At1g04180, At1g48910, At1g21430*) and transporter coding genes (*PIN1/2/3/4/5/6/7/8: At1g73590, At5g57090, At1g70940, At2g01420, At5g16530, At1g77110, At1g23080, At5g15100; AUX1/LAX1/2/3: At2g38120, At5g01240, At2g21050, At1g77690; ABCB1/4/19: At2g36910, At2g47000, At3g28860*). The displayed TF DEACoN results were narrowed down by setting the maximum *p*-value of 0.05 and logFC filter set on 2, corresponding to more than a fourfold enrichment.

APETALA2/ethylene-responsive element binding protein (AP2/EREBP) motifs in the promoter regions (up to 500 b.p. upstream of the start codon) of the auxin transporter genes (*PIN1, PIN2, PIN3*, *AUX1* and *LAX3*) and the promoters of the genes coding for the biosynthetic enzymes *TAA1*, *TAR1* and *TAR2* were identified by AthaMap [38].

### 4.5. Real-Time Quantitative RT-PCR Analyses

The real-time quantitative reverse transcription PCR analysis (qRT-PCR) of the *AUX1, LAX3, PIN1, PIN2* and *PIN3* gene expression was monitored in root-derived RNA samples (extracted with GeneJET Plant RNA Purification Kit Thermo Scientific, Basel, Switzerland) from Col-0, *ein2-1* and *ctr1-1* plants grown on vertical plates with ½ MS supplemented with 100 mM NaCl for 14 days. Complementary DNA (cDNA) was synthesized from 100 ng total RNA with iScript cDNA Synthesis Kit (Bio-Rad, Basel, Switzerland). The real time qRT-PCRs were performed with AccuPower^®^ GreenStar™ qPCR PreMix (Bioneer, Daejeon, South Korea) on three independent biological repeats and the analyses were carried out in three technical replicates of 10.0 μL reaction volumes with ‘PikoReal’ Real-Time PCR System (Thermo Scientific, Basel, Switzerland), at the following conditions: 95 °C for 15 min and 45 cycles of 95 °C for 10 s followed by 60 °C (for *LAX3*, *PIN1*, *PIN2* and *PIN3*) or 55 °C (for *AUX1*) for 30 s and final melting curves analysis with a temperature range of 60–95 °C in 0.2 °C increment for 60 s. The relative expression of the target genes was calculated using the 2^-ΔΔCq^ method [71] with two reference genes (*At3g18780*, *At5g60390*) for normalization of the relative quantification. Primers used in the qRT-PCRs are provided in Appendix A.

### 4.6. Statistical Analyses

The data in Figure 1 and Figure 4 were obtained from at least three independent experiments. Plants from three independent crosses with the respective reporters were analyzed to form three independent datasets. Each dataset contain measurements of at least 10 (up to 20) control or salt-treated individuals from the tested lines.

The assumed differences among the tested genotypes at control and salt stress conditions were analyzed by Student’s *t*-test (Figure 1) and one-way ANOVA (Figure 2, Figure 3, Figure 4, Figure 5 and Figure 6) using Excel software. Error bars on the graphs indicate standard deviation (SD) and the values were considered statistically significant at *p* ≤ 0.05.

## 5. Conclusions

Plant susceptibility/tolerance to salt stress is defined by multiple stress-responsive genes controlled by various signal transduction pathways. Our results demonstrate that ethylene signaling could also be engaged in salt stress response through the regulation of local auxin availability. This is sustained by the relative salt tolerance of *ctr1-1* mutation which is characterized by an altered ethylene/auxin regulatory loop leading to stabilized local auxin biosynthesis and polar transport. The salt hypersensitivity of the ethylene-insensitive mutant *ein2-1* could be due to the chronic root auxin deficiency.

## Figures and Tables

**Figure 1 plants-10-00452-f001:**
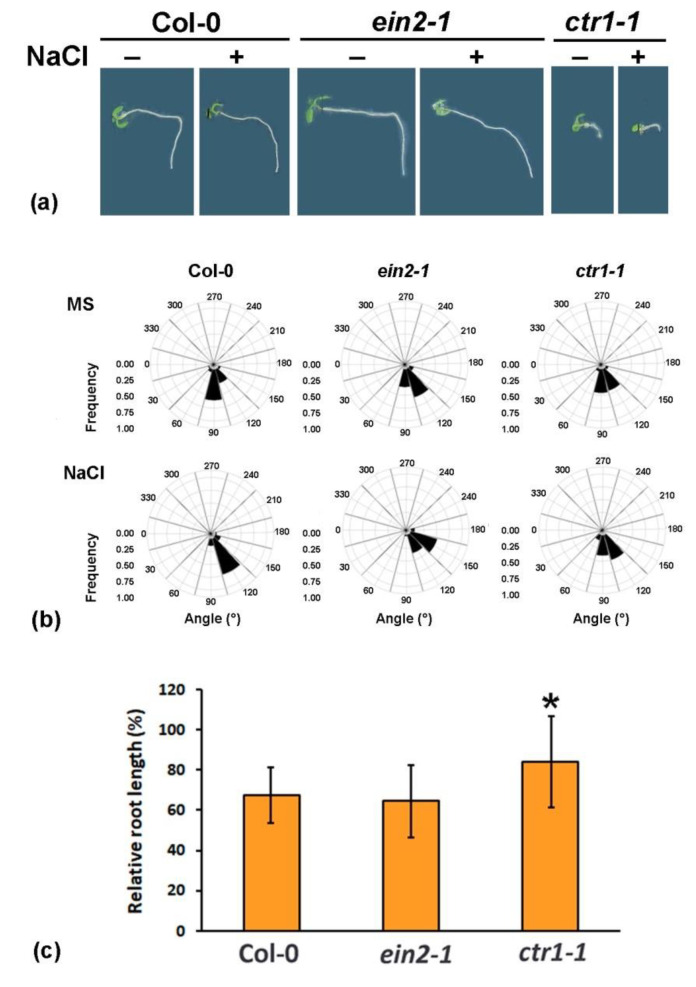
(**a**) Representative images of gravistimulated individuals with Col-0, *ein2-1* and *ctr1-1* genetic background on half-strength Murashige and Scoog (½ MS) media +/− 100 mM NaCl. (**b**) Quantification of root gravitropism of seedlings presented in (**a**). Roots were assigned to twelve 30 sectors on a gravitropism diagram. The size of each bar represents the frequency of seedlings showing the particular growth direction. (**c**) Root growth inhibition of the gravistimulated individuals with the wild type Col-0, and *ein2-1* and *ctr1-1* mutant backgrounds grown on ½ MS media +/− 100 mM NaCl. Asterisk represents statistically significant difference compared to NaCl-treated wild type (Col-0) (Student’s *t*-test, * *p* ≤ 0.05, *n* = 3 datasets, 20 individuals per experimental group were evaluated in each dataset).

**Figure 2 plants-10-00452-f002:**
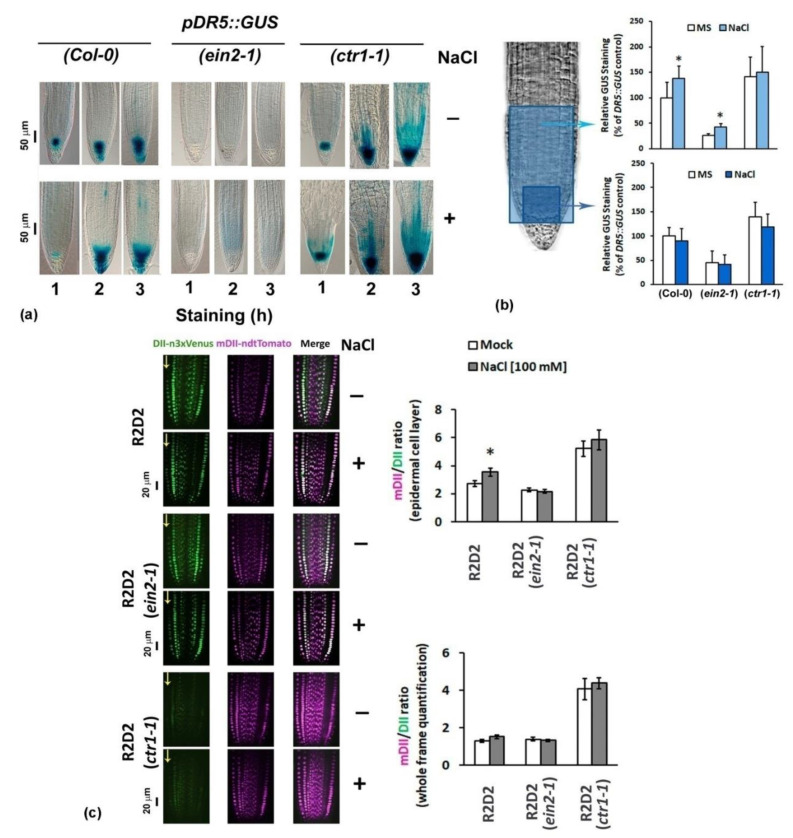
(**a**) Representative images of *pDR5::GUS*patterns in six-day-old control and 100 mM NaCl-treated (48 h) individuals with wild type Col-0, and *ein2-1* and *ctr1-1* mutant backgrounds (homozygous F3 plants), incubated for 1, 2 or 3 h in β-Glucuronidase (GUS) staining solution. (**b**) Whole frame and root meristem GUS intensity measured in 2 h-stained plants (Col-0, *ein2-1* and *ctr1-1*). Results are represented as percentage of the untreated control reporter. Asterisks depict statistically significant difference within the genotype (treated vs. non-treated), one-way ANOVA, *p* ≤ 0.05, *n* ≥ 3 datasets from independent crosses. **(c)** Fluorescent signal evaluation (whole frame and epidermis) in roots of control and 100 mM NaCl-treated (48 h) R2D2 plants and their homozygous F3 crosses with *ctr1-1* and *ein2-1*. Arrows mark the epidermal cell file. Asterisk depicts statistically significant difference within the genotype (treated vs. nontreated), *n* ≥ 10 individuals from one homozygous F3 line were analyzed.

**Figure 3 plants-10-00452-f003:**
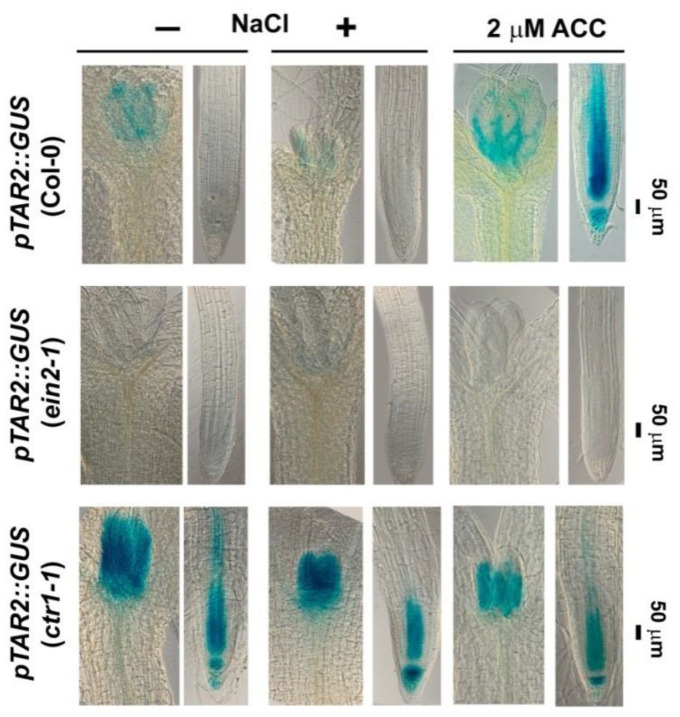
Staining patterns of *pTAR2::GUS* in roots and shoots of the wild type Col-0 and homozygous (F3) *ctr1-1* and *ein2-1* plants subjected to 100 mM NaCl treatment for 48 h. The *pTAR2::GUS* expression pattern in six-day-old plants grown on media supplemented with 2 µM 1-aminocyclopropane-1-carboxylic acid (ACC) is shown.

**Figure 4 plants-10-00452-f004:**
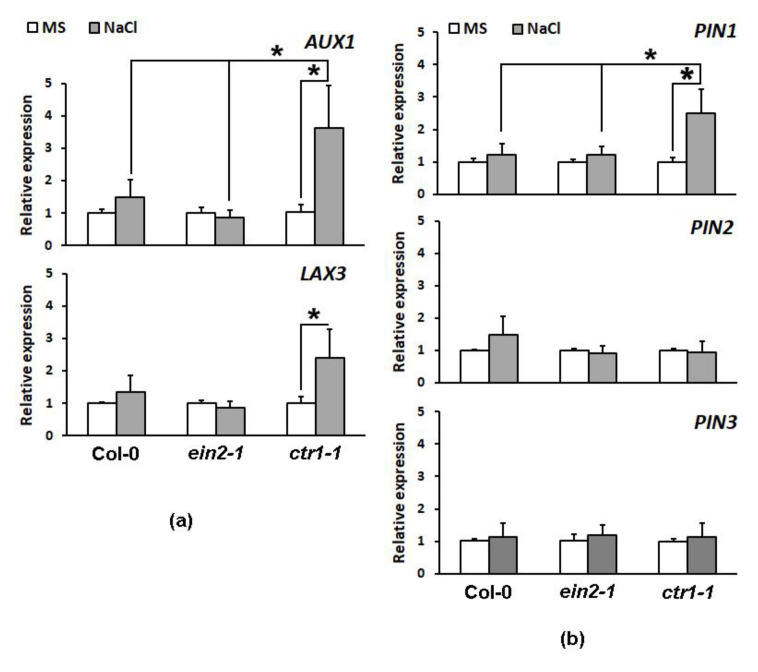
Relative expression of the auxin influx *AUX1* and *LAX3* (**a**), and efflux carrier genes *PIN1*, *PIN2* and *PIN3* (**b**) in the roots of the wild type Col-0, ethylene-insensitive *ein2-1*, and constitutive signaling mutant *ctr1-1* grown on ½ MS (white bars) and media supplemented with 100 mM NaCl (grey bars) at 14 DAG (day after germination). Asterisks depict significant differences compared to the salt-stressed wild type or compared to the control within the same genetic background (One-way ANOVA, *p* ≤ 0.05, *n* ≥ 3 datasets).

**Figure 5 plants-10-00452-f005:**
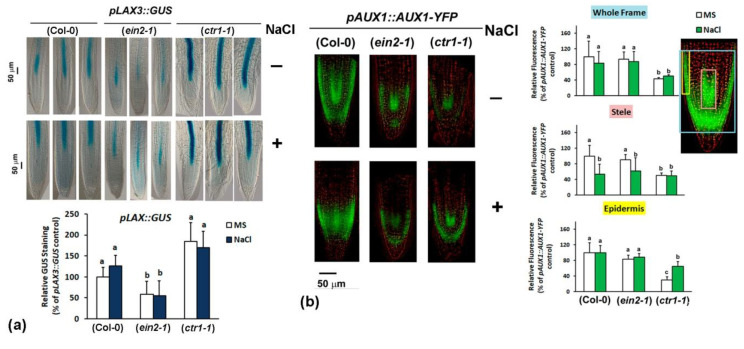
Auxin influx carrier expression in ethylene-signaling mutants subjected to treatment with 100 mM NaCl for 48 h: (**a**) GUS staining patterns and staining intensity measurement in roots of wild type Col-0 and homozygous F3 mutant *ctr1-1* and *ein2-1* plants carrying *pLAX3::GUS.* Three representative root images per line and per treatment are shown to demonstrate the consistency of the identified staining patterns. (**b**) Fluorescence intensity measurements in roots of *pAUX1::AUX1-YFP* reporters and their crosses with *ctr1-1* and *ein2-1* (homozygous F3 plants). The whole frame, stele, and epidermis macros used for the measurements are depicted on the root tip image in blue, pink and yellow, respectively. The letters in the graphs reflect significant differences in the different treatment groups found by comparison to the wild type background control (One-way ANOVA, *p* ≤ 0.05, and *n* ≥ 3 datasets from independent crosses).

**Figure 6 plants-10-00452-f006:**
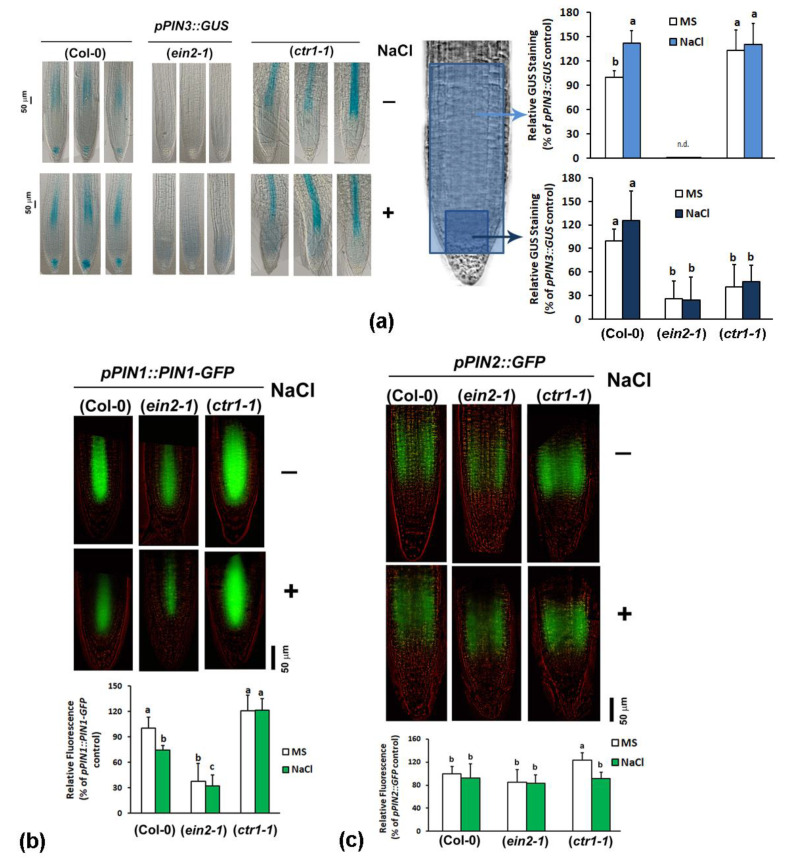
Auxin efflux carrier expression in ethylene-signaling mutants subjected to treatment with 100 mM NaCl for 48 h: (**a**) GUS staining patterns and staining intensity measurement in roots of wild type Col-0 and homozygous F3 mutant *ctr1-1* and *ein2-1* plants carrying *pPIN3::GUS* (“n.d.” in the graph means “not detected”). Three representative root images per line and per treatment are shown to demonstrate the consistency of the identified staining differences. The ‘whole frame’ and ‘root meristem’ macros used for the measurements are marked with light and dark blue, respectively. (**b**) Fluorescent intensity measurements (whole frame) in roots of *pPIN1::PIN1-GFP* reporters and their crosses with *ctr1-1* and *ein2-1* (homozygous F3 plants). (**c**) Fluorescent intensity measurements (whole frame) in roots of *pPIN2::PIN2-GFP* reporters and their crosses with *ctr1-1* and *ein2-1* (homozygous F3 plants). The letters in the graphs reflect significant differences in the different treatment groups which were found by comparison to the wild type background control. (One-way ANOVA, *p* ≤ 0.05, and *n* ≥ 3 datasets from independent crosses).

**Figure 7 plants-10-00452-f007:**
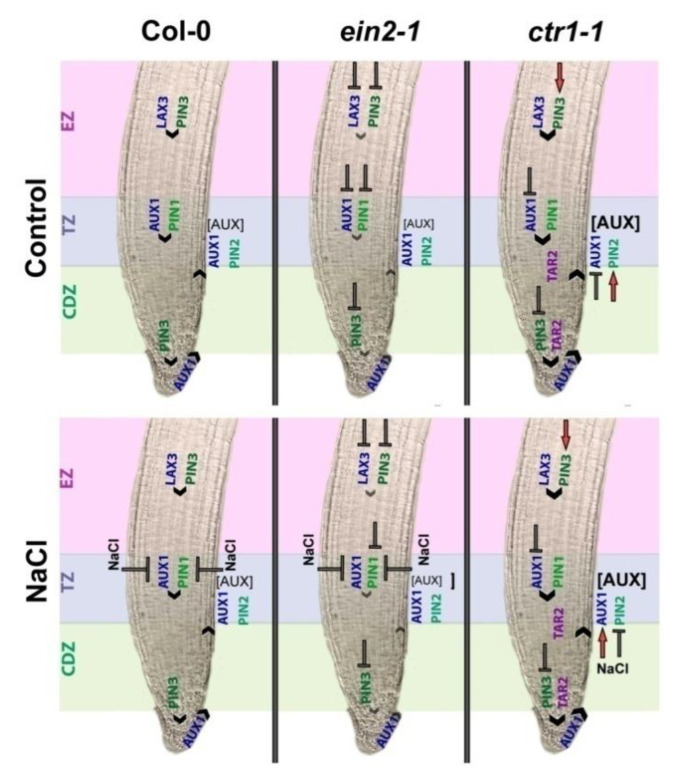
Model of root auxin homeostasis under salt stress in the Cell Division Zone (CDZ), Transzition Zone (TZ), and Elongation Zone (EZ) maintained by local auxin biosynthesis (TAR2) and polar transport (LAX3, AUX1, PIN1, PIN2, PIN3) in the wild type Col-0, *ein2-1* and *ctr1-1* Arabidopsis plants. The direction of auxin polar transport is depicted with black arrowheads. The T-shaped symbol marks inhibition and the red arrows designate activation compared to the wild type under the same conditions. The different font size of the letters in auxin concentration label [AUX] and the arrowheads reflects the variations between the wild type Col-0 and the mutant backgrounds.

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
