# Peer review of "The Diverse Salt-Stress Response of Arabidopsis ctr1-1 and ein2-1 Ethylene Signaling Mutants Is Linked to Altered Root Auxin Homeostasis"

_plants, 2021, doi:10.3390/plants10030452_

Round 1
Reviewer 1 Report
This manuscript ("The diverse salt stress response of Arabidopsis ctr1-1 and ein2-1 ethylene signaling mutants is linked to altered root auxin homeostasis") by Vaseva et al. exploited the ctr1-1 and ein2-1 mutants in the ethylene signaling pathway to explore the effects of salt stress on the ethylene-auxin crosstalk for root growth. The study is focused, combining the phenotyping of salt effects on growth gravitropic response and growth and the impact of salt on the expression of auxin biosynthetic genes and auxin transport genes. It uses the ctr1 mutant as a constitutive activated ethylene signaling mutant and ein2, in which ethylene signaling is blocked.
Salt stress inhibits root growth by increasing auxin in the cell division and transition zones (CDZ and TZ). It does so by activating the ethylene response. Ethylene is known to activate auxin biosynthesis. Maybe here, it can be mentioned that TAA1 is WEI8, identified from a mutant screen for weak ethylene insensitivity on root growth.
The ctr1-1 mutant with activated ethylene signaling has more produced auxin in the root, reduced root growth. And consequently is more resistant to salt stress. Salt by increasing slightly more auxin in ctr1 has little effects. Therefore, ctr1-1 root growth is not inhibited. One can ask whether this is not due to the reduced growth rate of the mutant (fig 1c).
The ein2 mutant with no ethylene signaling has nearly no auxin response in the root. It also impacts its salt sensitivity.
Then comes the question of selecting the reporter lines used for the analysis on the mutant lines. In my opinion, this is not justified. First, one would have expected to screen the effects of salt stress on all the report lines in WT. And then cross the ones showing a difference into the two mutant lines. For example, why TAR2? Why none of the YUCs? The effects of stress on TAA1, YUC8, and YUC9 have been well documented. For example, Liu et al., 2016, showed that EIN3 regulates YUC9 in response to aluminum stress.
The same goes true for the auxin transport genes. Why only AUX1 and LAX3?
Finally, there are discrepancies between the qPCR results and the reporter lines analysis. For example, in line 287, you indicated that "This observation is in line with the measured stronger salt-induced LAX3 expression in ctr1-1 roots detected by qRT-PCR". The qPCR saw an increase of LAX3 expression in ctr1-1 roots after salt treatment (figure 4a). But in Figure 5a, LAX3 is higher in the ctr1-1 roots, but no effects of salt are detected. Similar is true for PIN1 and PIN3. PIN3 increases in vascular tissue in ctr1 (salt or no salt), not in the qPCR. PIN1-GFP expression decreases by salt treatment in WT and ein2, but no changes in ctr1. But in the qPCR, PIN1 expression increase in ctr1-1 after salt treatment. This requires clarifications.
As a minor comment, IPA is used in line 193, but IPyA in line 198. I believe this is the same molecule, and only one acronyme should be used.
Author Response
We would like to thank the reviewer for the critical remarks and suggestions!
The text of the manuscript has been amended according to the recommendations of Reviewer 1 and Reviewer 2.
(Reviewer 3 had no major concerns and has recommended publication of the paper in its current form).
We have incorporated our answers bellow each of the remarks made by Reviewer 1.
“Reviewer 1
Comments and Suggestions for Authors
This manuscript ("The diverse salt stress response of Arabidopsis ctr1-1 and ein2-1 ethylene signaling mutants is linked to altered root auxin homeostasis") by Vaseva et al. exploited the ctr1-1 and ein2-1 mutants in the ethylene signaling pathway to explore the effects of salt stress on the ethylene-auxin crosstalk for root growth. The study is focused, combining the phenotyping of salt effects on growth gravitropic response and growth and the impact of salt on the expression of auxin biosynthetic genes and auxin transport genes. It uses the ctr1 mutant as a constitutive activated ethylene signaling mutant and ein2, in which ethylene signaling is blocked.
Salt stress inhibits root growth by increasing auxin in the cell division and transition zones (CDZ and TZ). It does so by activating the ethylene response. Ethylene is known to activate auxin biosynthesis.
Maybe here, it can be mentioned that TAA1 is WEI8, identified from a mutant screen for weak ethylene insensitivity on root growth.”
Answer: This suggestion was taken into account and the information has been added (Line 197): “TAA1, also known as Weak ethylene insensitive 8 (WEI8), has been initially identified from a mutant screen for weak ethylene insensitivity of root growth. [43].”
“The ctr1-1 mutant with activated ethylene signaling has more produced auxin in the root, reduced root growth. And consequently is more resistant to salt stress. Salt by increasing slightly more auxin in ctr1 has little effects. Therefore, ctr1-1 root growth is not inhibited. One can ask whether this is not due to the reduced growth rate of the mutant (fig 1c).”
Answer: The restricted growth is undeniably one of the first observable responses to stress and the higher auxin availability in the ctr1-1 root is in line with the low growth rate of the mutant. The higher auxin levels in ctr1-1 are likely linked to other responses which are under auxin/ethylene control and may as well contribute to the better performance of the mutant under challenging conditions. We studied some of the probable molecular mechanisms behind the relative salt tolerance of the constitutive mutant, particularly the ones linked to ROS scavenging and the SOS pathway, and we found some strong correlations with the ctr1-1 background. These results are described in another manuscript which is still under review.
“The ein2 mutant with no ethylene signaling has nearly no auxin response in the root. It also impacts its salt sensitivity.
Then comes the question of selecting the reporter lines used for the analysis on the mutant lines. In my opinion, this is not justified. First, one would have expected to screen the effects of salt stress on all the report lines in WT. And then cross the ones showing a difference into the two mutant lines.”
Answer: The preliminary assessment of the salt stress response of a number of auxin reporters revealed mostly diminishing signals, with certain variability depending on the evaluation time points. During these pilot tests, we also witnessed that the longer exposure to NaCl (more than 72 h) led to cell lyses in the root tips of the reporters which obstructed the visualization of the changes in the expression profiles. Particularly vulnerable in this regard were the GFP/YFP lines which showed cellular decay a bit earlier. Therefore, we limited the treatment in the experiments with the reporters to 48 h. Several important criteria for the selection of the auxin reporter lines were considered. One of them was the ethylene/ACC activation attesting to the ethylene signaling involvement in the control mechanisms (based on our previous experience with some of the used lines, as well as on published references). We also double-checked this particular feature by a detailed in silico characterization of the gene promoters for the availability of AP2/ERBP binding sites. We took into account the cell-type-specific expression of the efflux (PINs) and influx carriers (AUX, LAX) e.g. making sure that they cover both internal (stele) and the outermost cell files (epidermis, cortex) in the three zones of the root tip – the Cell Division Zone, the Transition Zone, and the Elongation Zone.
“For example, why TAR2? Why none of the YUCs? The effects of stress on TAA1, YUC8, and YUC9 have been well documented. For example, Liu et al., 2016, showed that EIN3 regulates YUC9 in response to aluminum stress.”
Answer: YUC8 indeed is both auxin- and ACC- inducible and its activity is strongly dependent on local polar auxin transport therefore it should have been a very good candidate. The same characteristics apply to TAR2 as well. Besides being proven to be actively involved in the ethylene-directed IAA biosynthesis (He et al 2011, Kriechbaumer et al. 2017), TAR2 has the additional priority to be operating immediately after TAA1 in the chain of enzymatic conversion of tryptophan to IAA and upstream of all YUCs. Another reason to consider TAR2 as a tool in the present study is that it shows a strong root-specific induction upon ACC treatment which is even more pronounced upon simultaneous application of auxin transport inhibitors (Brumos et al 2020, the Plant Cell). This suggests that TAR2 is a crosstalk point for ethylene and auxin signals. The other molecular player in the same branch of the auxin biosynthetic pathway TAR1 was not considered since it is weakly affected by ACC, which was unexpected providing the presence of EIN3 binding site in its promoter identified in the AthaMap platform. pYUC9:GUS reporter has a weaker response to ACC treatment therefore it was not considered for the genetic crosses with the ethylene mutants.
“The same goes true for the auxin transport genes. Why only AUX1 and LAX3?”
Answer: As already explained we have taken into account the two major characteristics of the reporters used in the study – ACC/ethylene regulation and activity site. Since the AthaMap analyses of the auxin transport gene promoters outlined the presence of ERF1 binding sites in pLAX3 and pPIN1 (as shown in Fig. S2) the respective reporter lines were considered as preferable candidates for the analyses of the ethylene signaling/auxin transport interactions. The choice of pAUX1::AUX1-YFP and pPIN2::PIN2-GFP was based on our previous experience with these reporters – we already have confirmed their ACC-responsiveness. We took into account the fact that AUX1 influx carrier operates both in the epidermis and LRC, as well as in the stele of CDZ. ProLAX3 conveniently adds-on coverage of the stele cell files above the TZ. The selection of the PIN reporters in regard to their cell-type-specific expression followed similar reasoning – pPIN1 and pPIN3 are expressed in the stele (CDZ and EZ/CDZ respectively) while pPIN2 is active in the outermost cell files of the root tip (epidermis/cortex).
We accept the reviewer’s criticism on the insufficiently explained choice of reporters and accordingly we included few extra lines clarifying their selection for the conducted research The following text has been added in Introduction (Line:98):
“The major criterion for the selection of the auxin reporter lines included in the study was their activation by 1-aminocyclopropane-1-carboxylic acid (ACC, the precursor of ethylene) attesting to the ethylene signaling involvement in the control mechanisms [37] and the presence of binding sites for APETALA2/Ethylene-responsive element binding proteins (AP2/ERBP) in the gene promoters identified by AthaMap analyses [38]. The cell-type specificity of the reporter’s expression was also considered in terms of sufficient coverage of the different cell files in the root tip zones.”
Additionally in the Suppl. Materials we also included the AthaMap profiles of the YUCs genes, which are now depicted below the already presented TAA1, TAR1, and TAR2 promoters (Fig S1).
“Finally, there are discrepancies between the qPCR results and the reporter lines analysis. For example, in line 287, you indicated that "This observation is in line with the measured stronger salt-induced LAX3 expression in ctr1-1 roots detected by qRT-PCR". The qPCR saw an increase of LAX3 expression in ctr1-1 roots after salt treatment (figure 4a). But in Figure 5a, LAX3 is higher in the ctr1-1 roots, but no effects of salt are detected. Similar is true for PIN1 and PIN3. PIN3 increases in vascular tissue in ctr1 (salt or no salt), not in the qPCR. PIN1-GFP expression decreases by salt treatment in WT and ein2, but no changes in ctr1. But in the qPCR, PIN1 expression increase in ctr1-1 after salt treatment. This requires clarifications.”
Answer: We would like to outline the differences between the treatment schemes of the two experimental approaches which are addressed in the section Materials and Methods. For the qPCRs we used roots harvested from plants continuously grown on 100 mM NaCl for 14 days. Due to the negative effect of NaCl on the root cell integrity, the work with the reporters was restricted to analyses of 7 days-old individuals initially grown under control conditions for 5 days and subsequently transferred on NaCl supplemented medium for only 48 h. Therefore a full overlap of the observed trends in both experimental models should not be expected. The advantage of the reporter systems is the cell-type-specific visualization of the expression of the particular gene which is not revealed by a conventional organ-specific qRT-PCR in which we extract total RNA from intact roots. However, both approaches sufficiently confirm the existing upregulation of local auxin transport caused by the ctr1-1 mutation: the reporters visualize the stable expression of the respective transport genes in the ctr1-1 background, and the qPCRs show that after prolonged cultivation on NaCl-containing medium ctr1-1 exhibits a relatively higher expression of some of the auxin transporter genes. To meet the requested clarification of the results obtained in the two experimental models we included the following text in section Results (2.4.3. Analyses of auxin transport reporters in ethylene mutant backgrounds):
“To address the effect of the ethylene signals on polar auxin transport in the roots subjected to salt stress with а cell-type-specific resolution we introduced auxin transport reporter constructs (pLAX3::GUS, pAUX1::AUX1-YFP, pPIN1::PIN1-GFP, pPIN2::PIN2-GFP, and pPIN3::GUS) into ctr1-1 and ein2-1 by crossing. The analyses were performed with 7 days-old homozygous (F3) plants subjected to 48 h cultivation on media containing 100 mM NaCl.”
As a minor comment, IPA is used in line 193, but IPyA in line 198. I believe this is the same molecule, and only one acronyme should be used.
Answer: We made the used abbreviation for indole pyruvate uniform throughout the text.
Reviewer 2 Report
In their manuscript „The diverse salt stress response of Arabidopsis ctr1-1 and ein2-1 ethylene signaling mutants is linked to altered root auxin homeostasis“, Vaseva et al. describe the auxin-related salt stress response of wild type plant and two well known ethylene signaling mutants. The choice of the two mutants was justified due to their altered gravitropic response on salt stress which was the starting point for this study. The authors use various molecular approaches (qPCR, various reporter genes, as well as in silico analysis) to corroborate their findings and making them convincing. The results point to the conclusion that, as previously described, auxin and ethylne responsens are interconected in a way that increased auxin steady-state levels in the root tip seem to provide a certain level of salt stress tolerance in the constitutive ethylene signaling mutant crt1-1.
In Fig.1 and Fig.2 the authors showed that the ctr1-1 mutant has significantly less impaired root elongation and gravitropism on 100 mM NaCl, in comparison to Col and ein2-1. This led to the question whether ethylene-dependent auxin distribution in the root might be responsible for that observation. Two reporter gene systems were apllied (Fig.2) to estimate auxin root tip levels: DR5-GUS and R2D2 fluorescence. With both reporters, a significant increase in the reporter signal was detected in wild type (Col) on prolonged salt treatment. ein2-1 showed different outcomes with different reporters (increase with GUS, uncanged with R2D2), whereas ctr1-1 showed increased GUS expression after 1 h on salt, but this efect was diminished after 3 hours. The results here are obviously not straightforward the conclusions should be drawn in relation to the exact time point and mutant, not generalised.
In Fig.3. it is rather convincing that on salt Col and ein2-1 decrease auxin biosynthesis (seen via the TAR-GUS reporter), whereas ctr1-1 keeps its auxin more or less stable. It is also clear that exogenous ethylene precursor ACC induces auxin biosynthesis in Col and ctr1-1, but not in ein2-1 as described previously. The authors corroborate this auxin-ethylene interplay by providing in silico evidence of several ethylene regulator binding site in promoters of auxin biosynthesis and transprt genes by AthaMAP and TF DEACoN analysis.
In Fig.4 qPCR experients demonstrate the upregulation of 2 influx carriers (AUX1, LAX3) and 1 efllux carrier (PIN2) on salt stress in the crt1-1 line.
In Fig.5 and Fig.6 the infllux carrier reporter and efflux carrier reporter genes were analysed in all 3 lines on salt in comparison Col. The result are somewhat ambivalent. Whereas the LAX3-GUS reporter is unafected by salt, the AUX1-YFP reporter is significantly upregulated only in the epidermis on salt. This observation follows the trend of increased auxin levels in ctr1-1 on salt. Further more, PIN3 and PIN1 remain the same on salt in ctr1-1, whereas PIN2 shows a slight decrease on salt in the same mutant. ctr1-1 is the most salt tolerant line, which is in correlation with increased levels of auxin (in some experiments), and increased AUX1-YFP expression. Other tested PIN proteins are either unafected (PIN3, PIN1) or decreased (PIN2).
Fig. 7 presents a fairly probable model trying to integrate the various findings of all experiments.
Despite the fact that the data are sometimes pointing in different directions, and are hard to interpert in one big picture, the strenght of this study is the rally relevant observation of constitutive ethylene signaling -dependent auxin upregulation in ctr1-1 which seems to act protective to the root elongation on salt stress.
Specific comments:
Please, justify the 100mM NaCl concentration used in experiments. Did you make some preliminary tests with different NaCl concentrations and then chose the 100mM value? I miss a short explanation how you came to 100 mM.
Lanes 174-175: this is a rather generalized sentence adresing an experiment that didn't have straightforward outcomes (Col reacted differently after 1h vs. 3h, ctr1-1 also has that difference 1h vs. 3h). So, I expect here a statement that is more in line with the actual results. You could specify to which part of the experiment you reffer.
Lane 177: „These observations were confirmed…“ Please, specify which observations? The reader does not follow which particular observations you are reffering to. The statement is to general since all 3 lines do not respond all in the same way/direction. Be more specific with the conclisions.
Lane 302-303: statistical significance of treated vs. untreated samples would be of more importance. The difference WT vs. mutants is visible on GUS stained roots. The more important observation is that on salt the auxin infllux, as visualised form the reporter gene, seems to be unaffected by higher ethilene signaling.
Author Response
We would like to thank the reviewer for the critical remarks and suggestions!
The text of the manuscript has been amended according to the recommendations of Reviewer 1 and Reviewer 2.
(Reviewer 3 had no major concerns and has recommended publication of the paper in its current form).
We have incorporated our answers bellow each of the remarks made by Reviewer 2.
"Reviewer 2
“In their manuscript „The diverse salt stress response of Arabidopsis ctr1-1 and ein2-1 ethylene signaling mutants is linked to altered root auxin homeostasis“, Vaseva et al. describe the auxin-related salt stress response of wild type plant and two well known ethylene signaling mutants. The choice of the two mutants was justified due to their altered gravitropic response on salt stress which was the starting point for this study. The authors use various molecular approaches (qPCR, various reporter genes, as well as in silico analysis) to corroborate their findings and making them convincing. The results point to the conclusion that, as previously described, auxin and ethylne responsens are interconected in a way that increased auxin steady-state levels in the root tip seem to provide a certain level of salt stress tolerance in the constitutive ethylene signaling mutant crt1-1.
In Fig.1 and Fig.2 the authors showed that the ctr1-1 mutant has significantly less impaired root elongation and gravitropism on 100 mM NaCl, in comparison to Col and ein2-1. This led to the question whether ethylene-dependent auxin distribution in the root might be responsible for that observation. Two reporter gene systems were apllied (Fig.2) to estimate auxin root tip levels: DR5-GUS and R2D2 fluorescence. With both reporters, a significant increase in the reporter signal was detected in wild type (Col) on prolonged salt treatment. ein2-1 showed different outcomes with different reporters (increase with GUS, uncanged with R2D2), whereas ctr1-1 showed increased GUS expression after 1 h on salt, but this efect was diminished after 3 hours. The results here are obviously not straightforward the conclusions should be drawn in relation to the exact time point and mutant, not generalised.”
Answer: In line with the made remark we revised the text narrowing down the conclusions by reflecting only the concrete observations (see lines 161, 174-175; 177, 187).
“In Fig.3. it is rather convincing that on salt Col and ein2-1 decrease auxin biosynthesis (seen via the TAR-GUS reporter), whereas ctr1-1 keeps its auxin more or less stable. It is also clear that exogenous ethylene precursor ACC induces auxin biosynthesis in Col and ctr1-1, but not in ein2-1 as described previously. The authors corroborate this auxin-ethylene interplay by providing in silico evidence of several ethylene regulator binding site in promoters of auxin biosynthesis and transprt genes by AthaMAP and TF DEACoN analysis.
“In Fig.4 qPCR experients demonstrate the upregulation of 2 influx carriers (AUX1, LAX3) and 1 efllux carrier (PIN2) on salt stress in the crt1-1 line.”
A clarification: The reviewer probably had in mind PIN1, not PIN2.
“In Fig.5 and Fig.6 the infllux carrier reporter and efflux carrier reporter genes were analysed in all 3 lines on salt in comparison Col. The result are somewhat ambivalent. Whereas the LAX3-GUS reporter is unafected by salt, the AUX1-YFP reporter is significantly upregulated only in the epidermis on salt. This observation follows the trend of increased auxin levels in ctr1-1 on salt. Further more, PIN3 and PIN1 remain the same on salt in ctr1-1, whereas PIN2 shows a slight decrease on salt in the same mutant. ctr1-1 is the most salt tolerant line, which is in correlation with increased levels of auxin (in some experiments), and increased AUX1-YFP expression. Other tested PIN proteins are either unafected (PIN3, PIN1) or decreased (PIN2).
Fig. 7 presents a fairly probable model trying to integrate the various findings of all experiments.
Despite the fact that the data are sometimes pointing in different directions, and are hard to interpert in one big picture, the strenght of this study is the rally relevant observation of constitutive ethylene signaling -dependent auxin upregulation in ctr1-1 which seems to act protective to the root elongation on salt stress.
Specific comments:
Please, justify the 100mM NaCl concentration used in experiments. Did you make some preliminary tests with different NaCl concentrations and then chose the 100mM value? I miss a short explanation how you came to 100 mM.”
Answer: We have tested the effect of increasing salt concentrations (0-50-100-150-200 mM) on the germination and found that the wild type Col-0 tolerated salinity of the substrates up to 100-150 mM NaCl. The constitutive signaling mutant ctr1-1 was capable to germinate even at the inhibiting concentration of 200 mM NaCl. However the ethylene insensitive ein2-1 mutant had delayed germination already at 100 mM NaCl. The performed survival test done with the same NaCl concentrations demonstrated that the number of ein2-1 growing individuals at 4 DAG was comparable to the ones of the wild type and the ethylene constitutive mutant ctr1-1, thus revealing that the ethylene insensitive mutant was capable to compensate the delay from the germination stage. However, the survival test showed that ein2-1 individuals were severely affected by 150 mM NaCl showing only 20% survival rates. Therefore, we chose to perform our analyses with 100 mM NaCl which is a threshold concentration tolerated by the three analyzed genetic backgrounds.These results (germination and survival test) are part of another paper which was recently submitted and is still under review. We have addressed the remark by clarifing the choice of the NaCl concentration in section Materials and Methods (4.1. Plant material and growth conditions, second paragraph):
“The salt stress was imposed by adding 100 mM NaCl to the media as this was identified as a threshold concentration at which the three analyzed genetic backgrounds (Col-0, ein2-1 and ctr1-1) had comparable germination and survival rates (unpublished results).”)
“Lanes 174-175: this is a rather generalized sentence adresing an experiment that didn't have straightforward outcomes (Col reacted differently after 1h vs. 3h, ctr1-1 also has that difference 1h vs. 3h). So, I expect here a statement that is more in line with the actual results. You could specify to which part of the experiment you reffer.”
Answer: We thank the reviewer for the remark! Following the recommendation we have amended the text as follows: “Therefore, the detected salt stress-related increase in auxin signaling after 2 hours of GUS staining should be assigned mostly to the cell types outside this part of the root tip”.
“Lane 177: „These observations were confirmed…“ Please, specify which observations? The reader does not follow which particular observations you are reffering to. The statement is to general since all 3 lines do not respond all in the same way/direction. Be more specific with the conclisions.”
Answer: The text was rephrased to clarify the statement and now it reads: ” The assumption that the increased auxin activity provoked by the salt stress should be assigned to the cells outside the root meristem was confirmed by the measurement of the fluorescent signal in the NaCl-treated highly sensitive auxin sensor R2D2 (Figure 2c).”
“Lane 302-303: statistical significance of treated vs. untreated samples would be of more importance. The difference WT vs. mutants is visible on GUS stained roots. The more important observation is that on salt the auxin infllux, as visualised form the reporter gene, seems to be unaffected by higher ethilene signaling.”
Answer: We used letters in the graph to reflect the differences among the experimental groups, taking into account both the genetic backgrounds and the treatment and this was done by comparing each one to the control wild type situation. The text was amended to reflect the reviewer’s critical remark and now it reads: “The letters in the graphs reflect significant differences in the different treatment groups which were found by comparison to the wild type background control.” The same text was included at the end of Figure 6 caption as well.
Reviewer 3 Report
In this manuscript, the author explained the diverse salt stress response of Arabidopsis ctr1-1 and ein2-1 ethylene signaling mutants and linked it to the altered root auxin homeostasis. In this study, the author studies the effect of salt in ethylene sensitive mutant ein2-1 and constitutive mutant ctr1-1. The continuously operating ethylene signaling pathway in ctr1-1 was linked to increased auxin biosynthesis. This is a significant study in the context of stress tolerance interplay with ethylene signaling. The manuscript is very well written, and the experiments are precise and cover all aspects. I found no major flaws in this manuscript. Hence I recommend this manuscript to be accepted in its current format.
Author Response
We would like to thank the reviewer for the positive evaluation of our work!
Round 2
Reviewer 1 Report
The authors argumentatively answered my comments. I think all concerns have been answered on my satisfaction.